# Cu&Si Core–Shell Nanowire Thin Film as High-Performance Anode Materials for Lithium Ion Batteries

Lifeng Zhang [1], Linchao Zhang [1,*], Zhuoming Xie [1] and Junfeng Yang [1,2,*]

1   Key Laboratory of Materials Physics, Institute of Solid State Physics, Hefei Institute of Physical Science, Chinese Academy of Sciences, Hefei 230032, China; zlf05358@mail.ustc.edu.cn (L.Z.); zmxie@issp.ac.cn (Z.X.)
2   Lu'an Branch, Anhui Institute of Innovation for Industrial Technology, Lu'an 237100, China
*   Correspondence: lczhang@issp.ac.cn (L.Z.); jfyang@issp.ac.cn (J.Y.)

**Abstract:** Cu@Si core–shell nanowire thin films with a $Cu_3Si$ interface between the Cu and Si were synthesized by slurry casting and subsequent magnetron sputtering and investigated as anode materials for lithium ion batteries. In this constructed core–shell architecture, the Cu nanowires were connected to each other or to the Cu foil, forming a three-dimensional electron-conductive network and as mechanical support for the Si during cycling. Meanwhile, the $Cu_3Si$ layer can enhance the interface adhesion strength of the Cu core and Si shell; a large amount of void spaces between the Cu@Si nanowires could accommodate the lithiation-induced volume expansion and facilitate electrolyte impregnation. As a consequence, this electrode exhibits impressive electrochemical properties: the initial discharge capacity and initial coulombic efficiency is 3193 mAh/g and 87%, respectively. After 500 cycles, the discharge capacity is about 948 mAh/g, three times that of graphite, corresponding to an average capacity fading rate of 0.2% per cycle.

**Keywords:** lithium ion battery; anode; silicon; core–shell structure; magnetron sputtering

## 1. Introduction

Silicon (Si) has the highest known theoretical capacity of 4200 mAh/g, almost ten times that of the currently used graphite anode, and is therefore being considered as the most promising anode material for high-energy-density lithium ion batteries (LIBs) [1,2]. Unfortunately, Si suffers from as high as 300% volume change upon full (de)lithiation, inducing pulverization of the silicon particles and therefore losing electrical connectivity from the current collector and the thickening of the solid electrolyte interface (SEI) layer upon cycling; these eventually hinder the practical application of Si-based anode materials in LIBs [3,4].

To tackle these issues, enormous attention has been directed towards the development of Si nanostructured materials [5], of which Si nanowires (SiNWs) have attracted considerable interest owing to its small lithium diffusion length and facile strain relaxation during (de)lithiation [6,7]. However, the complete lithiation of Si nanowire impedes charge transport in the longitudinal direction, limiting its rate performance [8]. Cui et al. successfully synthesized a core–shell structured Si nanowire with a crystallite Si (c-Si) core and amorphous Si (a-Si) shell, in which the a-Si shell can be cycled alone as lithium ion storage whereas the c-Si core remains intact as mechanical sturdy support and efficient electron transport pathways by setting an appropriate cut-off potential owing to the higher lithiation potential of a-Si than c-Si ($\sim$220 mV vs. $\sim$120 mV, respectively), resulting in significant improvement in electrochemical performance over traditional Si nanowires, such as a high charge capacity of 1000 mAh/g and a capacity retention of 90% over 100 cycles [9]. On the heels of this work, a series of core–shell nanowires with Si as the core and electron conductive material as shell [10,11], or with Si as the shell and other electron conductive materials as the core [12,13], also have been fabricated, contributing to great achievement in Si nanowire-based anodes. However, they not only involved high temperature, complex

steps and the use of catalysts during preparation, but also have a low tap density when applied as an anode. Therefore, the Si core–shell nanowire-based anodes are relatively expensive to produce and hard to scale up.

Alternatively, Si film anodes have also been studied since they are binder and conductive additive free. Ab-initio calculations suggested that the critical thickness of the silicon film to avoid crack is about 100 nm, beyond which the silicon film would fracture and lead to rapid capacity fading [14]. Such a thin Si-film is not able to provide sufficient Si loading density for practical application [15]. By using a Si-based multilayer thin film, with alternating layers of inactive materials as a buffer layer [16,17], its loading density can be increased but only to a very limited extent.

In this study, to combine the advantage of both a Si thin film and core–shell Si nanowire, Cu@Si core–shell nanowire thin film (Cu&Si CSNWF) electrodes were fabricated through slurry casting, heat treatment, and magnetron sputtering. Subsequently, their electrochemical properties were investigated in detail. In comparison with other core materials, the Cu nanowire core has obvious advantage such as (1) higher electrical conductivity and fracture toughness [18]; (2) it is inactive to lithium and thus experiences no volume change during the lithiation/delithiation of Si; and (3) the Cu nanowires we used are commercially available and very cheap in price, and easy to be scale up in practical application.

## 2. Materials and Methods

### 2.1. Cu@Si Core–Shell Nanowire Thin Film (CSNWF) Preparation

Cu nanowires were purchased from Hongwu Nanometer Material CO., LTD.,Xuzhou China. Cu&Si CSNWFs were prepared by three steps as schematically illustrated in Figure 1. Firstly, 2 mg Cu nanowires (CuNWs), as received without any pretreatment, were uniformly dispersed into 10 mL isopropanol (IPA), and then the slurry was casted onto 50 μm-thick Cu foil, followed by heating at 280 °C for 2 h under the reductive atmosphere of 5 v% H2 + 95 v% Ar to evaporate IPA and reduce the oxide on the surface of the CuNWs. As a result, the Cu foil was coated with a layer of cross-linked CuNW sediment. Finally, the Si layer was deposited onto them without intentional heating or cooling to obtain Cu@Si core–shell nanowire thin films by direct current magnetron sputtering of the Si target. The sputtering power is about 120 W, the distance of the target to the substrate is about 50 mm and the work pressure is about 0.4 Pa. For comparison, pure Si film was simultaneously deposited on Cu foil and glass substrate, respectively, under the same sputtering parameter. The thickness of the pure Si films is determined as 456 nm through observing its cross-section thickness using SEM. The loading density of the as-deposited pure Si thin films was calculated under the assumption that the density of Si thin film is 2.33 g cm$^{-3}$, as reported in literature [19], which is the highest density achievable for the as-deposited films and will give the most conservative estimate for the specific capacity of our electrode materials.

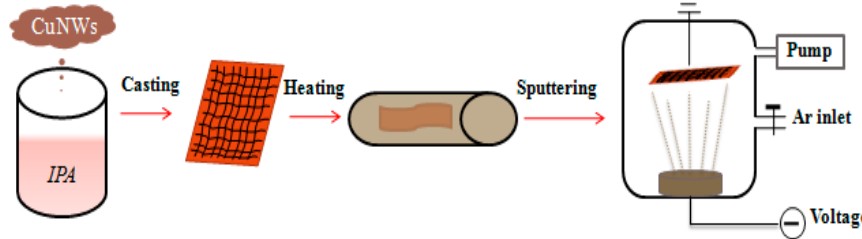

**Figure 1.** Schematic diagram of the preparation procedure for the Cu@Si core–shell nanowire thin films (CSNWFs).

### 2.2. Characterization and Electrochemical Measurement

The crystal structure of the Cu nanowire and as-deposited pure Si and Cu&Si CSNWF were characterized, respectively, by X-ray diffraction (XRD, Rikagu Smart Lab) with Cu Kα radiation (λ = 1.541 Å). The surface morphology and structure of the films were observed

by field-emission gun scanning electron microscopy (FEG-SEM, Zeiss Ultra-Plus) and scanning transmission electron microscopy (STEM, FEI Titan). The samples for STEM were made by the dual-beam focused ion beam (FIB) technique. A Swagelok-type two-electrode cell was constructed in an argon-filled glove box with a high-purity metallic lithium disk (Ø = 12 mm) as the counter electrode and reference electrode, Whatman filter paper (Ø = 13 mm) as the separator, and the Cu&Si core–shell nanowire thin film on copper foil (Ø = 10 mm) as the working electrode, without addition of any binder and conductive additive. The electrolyte was 1 mol $L^{-1}$ LiPF6 in a 1:1 (volume ratio) mixture of ethylene carbonate (EC) and dimethyl carbonate (DMC). Electrochemical impedance spectroscopy (EIS) and cyclic voltammetry (CV) tests were carried out on a PAR Versastat-2 electrochemistry system. EIS was measured over the frequency range from 0.01 to about 106 Hz with an AC amplitude of 10 mV. Cyclic voltammetry (CV) was recorded in the voltage range of 0.01–1 V (vs. Li/Li+) at a scan rate of 0.2 mV $s^{-1}$. Discharge/charge capacities were tested in an Arbin BT 2000 multichannel battery tester by galvanostatic cycling the half cells over the potential range between 0.01 and 1.5 V vs. Li/Li+ at a 0.1 C rate. C rates were calculated on the basis of a theoretical capacity of 4200 mAh/g. After the cycling test, the working electrode was taken out through disassembling the Swagelok-type cell at first and then rinsing it in dimethyl carbonate (DMC) and acetone, finally dried, and subsequently put in a glove box for further examination. All the electrochemical properties were measured at room temperature.

## 3. Results and Discussion

### 3.1. Characterization of Cu@Si CSNWFs

Figure 2a shows the XRD pattern of the as-received Cu nanowires, Cu nanowire coated Cu foil after thermal treatment, and Cu&Si CSNWFs. The diffraction patterns of the as-received Cu nanowire can be well assigned to Cu and Cu oxides (CuO and $Cu_2O$) [20], suggesting the oxidation of the Cu nanowire surface during storage. After heat treatment at 280 °C for 2 h in the reductive atmosphere of 5 v.% H2 + 95 v.% Ar, diffraction peaks corresponding to the Cu oxides disappear completely, indicating that this heat treatment is an effective way to remove Cu oxides. After heat treatment and Si deposition, four new peaks are observed; these peaks match well with Si and $Cu_3Si$ [21–24], respectively, demonstrating that the sputter-deposited Si is crystalline and there is formation of the crystalline $Cu_3Si$ phase. It is noteworthy that Chen et al. also found the formation of $Cu_3Si$ during sputtering of a Cu layer on the surface of Si nanowires [25]. Figure 2b,c exhibit the SEM images of the Cu nanowire-coated Cu foil before and after Si deposition, respectively. It is clearly seen that the Cu nanowires connect either to each other or to the Cu foil substrate, forming a continuous electron-conductive pathway. After Si deposition, there is an apparent increase in diameter of the Cu nanowires, and meanwhile a large number of void spaces could be observed in between them (Figure 2c). Based on Figure 2b,c, the corresponding diameter distribution of the Cu nanowire before and after Si deposition was statistically analyzed and exhibited in Figure 2d,e. The diameter of the Cu nanowire ranges from 50 to 300 nm, mainly between 150 and 200 nm. After Si deposition, the diameter is increased to 300–650 nm, and mainly lies in the range of 400–500 nm, suggesting the formation of a Cu@Si core–shell nanowire. The formation of voids was ascribed to the shadowing effect of the Cu nanowires during Si deposition. Figure 2f exhibits the STEM image of an individual core–shell nanowire, in which the core and shell can be distinguished by the obvious contrast in the image. The brighter inner part is the Cu core, which is covered by a continuous uneven gray Si shell. In order to further confirm the elemental composition, we have performed an EDS test in the mode of linear scanning along the straight line from A to B, as shown in Figure 2g. It can be recognized that this single Cu@Si core–shell nanowire has a Cu core of ~200 nm in diameter and a Si shell of 125 nm in thickness, with only trace amounts of oxygen. Furthermore, an obvious mixed layer of Cu and Si is also observed, corresponding to a thin layer of $Cu_3Si$, as confirmed by the XRD in Figure 2a, which is

beneficial for the enhancement of adhesion strength between the Cu core and Si shell. Based on the above, it is concluded that the Cu@Si CSNWFs were successfully fabricated.

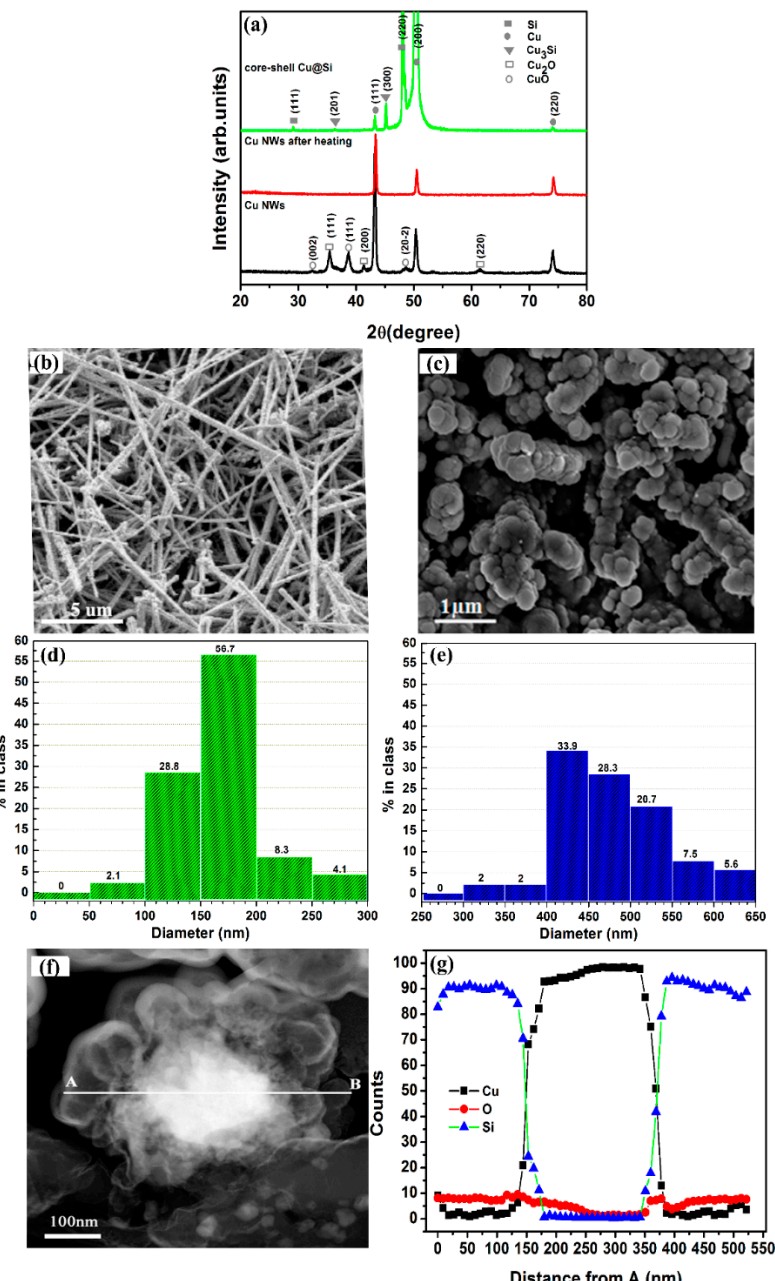

**Figure 2.** (**a**) XRD pattern of the as-received Cu nanowire on a Cu foil substrate before and after heat treatment at 280 °C for 2 h in a reductive atmosphere of 95 v.% Ar + 5 v.% H2, with and without Si deposition; (**b**) SEM image of the Cu nanowire (NW)-coated Cu foil substrate after heating treatment and the corresponding size distribution of the diameter of the Cu nanowire before (**b**,**d**) and after (**c**,**e**) Si deposition; (**f**) STEM image of a single Cu@Si core–shell nanowire and (**g**) its composition analysis by line scanning EDS along the straight line.

### 3.2. Electrochemical Performance of Cu@Si CSNWFs

Figure 3a exhibits the charge/discharge curves of the Cu@Si CSNWF electrode after the 1st, 2nd, 5th, and 10th cycle at a current rate of 0.1 C between 0.01 and 1.5 V. The potential window is 0.23–0.01 V during discharging and 0.23–0.55 V during charging, respectively. The voltage profiles exhibit a typical sloping and smooth curve without appearance of a distinct potential plateau. Figure 3b demonstrates the specific discharge capacity and

charge efficiency of the Cu@Si CSNWFs electrode as a function of cycle number. The initial discharge capacity is 3193 mAh/g. After 500 cycles, the discharge capacity is still as high as 948 mAh/g, corresponding to an average capacity fading rate of only 0.2% per cycle. As for the charge efficiency, the initial efficiency is close to 87%, and then quickly increases and lies between 96% and 99.8% during subsequent cycling. These suggest a good cycling stability of the Cu@Si CSNWF electrodes. In contrast, the specific discharge capacity of 456 nm-thick pure Si films fades rapidly with cycling, which is typical for a thick Si film electrode [26]. The higher initial charge efficiency and better cycling performance of Cu@Si CSNWF electrodes than pure Si film could be attributed to the improved electric conductivity and special core–shell structure. The electrical conductivity of $Cu_3Si$ was about $2 \times 10^4$ S cm$^{-1}$, a little lower than the Cu nanowire but much higher than pure silicon of about 10–5 S cm$^{-1}$ [27]. Indeed, it has been reported that increasing the electronic conductivity of Si films could improve the initial coulombic efficiency significantly [28,29]. Moreover, the $Cu_3Si$ layer between the Si shell and Cu core functions as an adhesive layer, and could enhance the adhesion strength between the Cu nanowire core and Si shell and therefore improve the cycling stability of the Cu@Si CSNWF electrodes in spite of a little sacrifice in capacity due to the electrochemical inactivity of $Cu_3Si$ to Li [24]. Kim et al. [30] also attributed the enhanced cycling stability of a Si–Cu–graphite composite to the formation of copper silicide in the interface between Cu and Si. Figure 3c exhibits the rate capability of the Cu@Si CSNWF electrode. The discharge capacity is about 2060 mAh/g at 0.2 C and exhibits a decreasing trend with the increase in current density. As the current density is increased to 0.5 C, 1.5 C, 3 C, and 6 C, the discharge capacity dropped down to ~1900 mAh/g, ~1600 mAh/g, ~1200 mAh/g, and~700 mAh/g, respectively. When the current density was returned to 0.2 C, the discharge capacity was increased to 2025 mAh/g, almost the same value as that in the beginning, demonstrating a good rate capability of our Cu@Si CSNWF electrodes. In addition, the coulombic efficiency at different current densities is beyond 90%, with the exception of the first cycle at 0.2 C, 3 C, and 6 C rates, further confirming the good rate performance of the Cu@Si CSNWF electrodes.

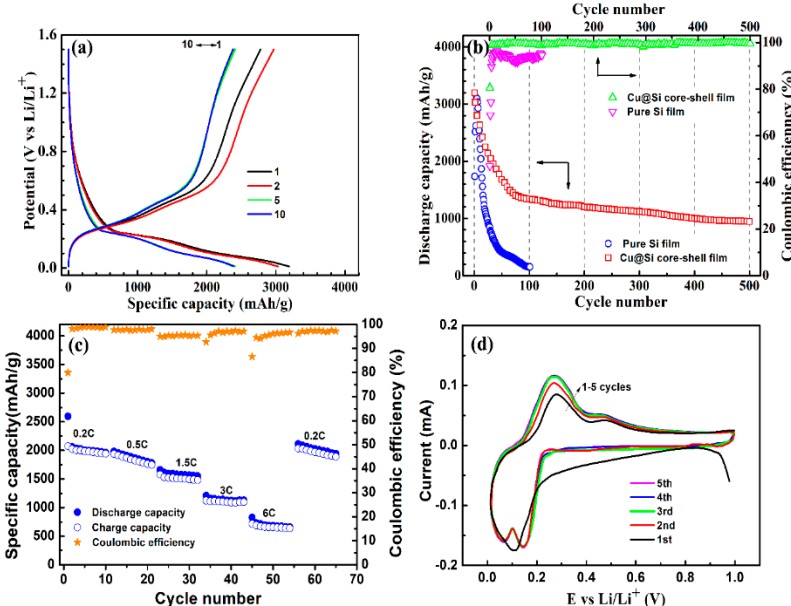

**Figure 3.** (**a**)Voltage profile of the Cu@Si CSNWFs at the 1st, 2nd, 5th, and 10th cycle at a current rate of 0.1 C between 0.01 and 1.5 V; (**b**) the specific discharge capacity and coulombic efficiency of the Cu@Si CSNWFs and pure Si films as a function of cycle number at a rate of 0.16 C between 0.01 and 1.5 V; (**c**) the specific discharge capacity of Cu@Si CSNWFs as a function of the cycle number and current rate; (**d**) cyclic voltammetry curves of the Cu–Si core–shell nanowires embedded Si thin films (CSNEFs) in the potential range of 0.01–1.2 V at a scan rate of 0.2 mV s$^{-1}$.

The good rate performance can be understood by the fast and efficient electron transfer of the Cu nanowire core and by the decrease in diffusion length of the lithium ions, as schematically illustrated in Figure 4, in that the diffusion length of the lithium ion in pure Si films is much longer than in our Cu@Si CSNWF. Figure 3d exhibits CV curves for the Cu@Si CSNWF electrode in the potential range of 0.01–1.2 V at a scan rate of 0.2 mV s$^{-1}$. It is clearly seen that the CV curve of the first cycle is distinct from those of subsequent cycles, especially for the cathodic branch. In the first cathodic scan, the current increases gradually as the potential is decreased from 0.76 V to around 0.23 V, which is ascribed to the formation of a solid electrolyte interface (SEI) film [31,32]. Thereafter, it starts to increase rapidly, leading eventually to the formation of a broad peak between 0.05 V and 0.20 V and corresponding to the lithiation of Si. In the first anodic scan, a distinct peak at 0.31 V and a weak and broad peak at 0.48 V were observed, which could be assigned to the transformation between the Li–Si alloy to Si. Since the second cycle, the cathodic current was kept almost unchanged in the potential range of 0.72 to 0.23 V; the current is substantially lower than that in the first cycle. These data imply that the electrolyte decomposition during the second cycle is substantially suppressed. Instead, two new peaks located at 0.15 and 0.05 V appear, which could be assigned to the formation of the Li–Si alloy [33]. As of the third cycle, the cathodic and anodic current peaks settled rapidly and maintained a stable pattern, eventually resulting in the peaks superimposing on each other, suggesting a good cycling performance in the following cycles. In contrast, Cui et al. reported that the current peak intensities increased gradually in the first few to ten cycles for the Si nanowire electrodes because the more active materials are activated to react with the lithium during cycling [7,34]. These results indicate that there is no gradual activation process of the active materials in our electrode; i.e., all active materials can be fully lithiated within two cycles, owing to the superior electrical conductivity of the Cu core in the core–shell structure and the void space for electrolyte impregnation, therefore reducing the diffusion length of the lithium ion.

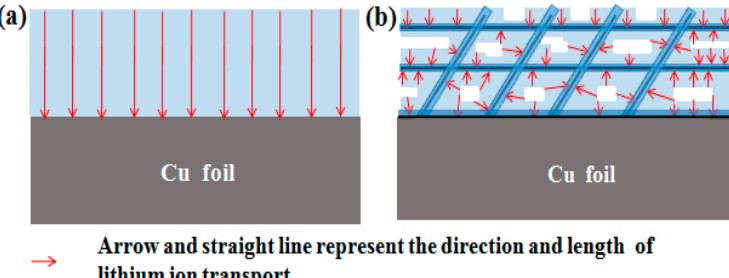

**Figure 4.** (**a**,**b**) Schematic diagram of the lithium ion diffusion in pure Si film and CSNEFs, respectively.

In order to clarify the mechanism of the enhanced electrochemical performance of the Cu@Si CSNWF in comparison with pure Si films, EIS, SEM, and STEM measurements were performed after the cycling test. Figure 5a,b exhibit the EIS spectra of the Cu@Si CSNWFs and pure Si films, respectively, after the cycling test in the state of full de-lithiation. As can be seen, for both electrodes the obtained Nyquist plots consisted of a high-frequency (HF) depressed semicircle and a low-frequency (LF) inclined line. It is accepted that in the fully delithiated state, the diameter of the HF-depressed semicircle is mainly ascribed to the intrinsic electronic resistance and the contact resistance (materia–material and material– current collector) while the LF inclined line is attributed to the lithium ion diffusion within the electrode [35,36]. The diameter of the HF semicircle for the pure Si film electrode is about 310 Ω after the first cycle, larger than that of the Cu@Si CSNWF (about 100 Ω), indicating that the electronic conductivity of the Cu@Si CSNWFs is much better than pure Si films. As the cycle proceeds, the HF semicircle diameter for the Cu@Si CSNWF increase to about 110 Ω after 30 cycles whereas that for pure Si films is increased up to about 530 Ω after 20 cycles, demonstrating the better microstructure stability of the Cu@Si CSNWFs than pure Si films. Figure 6a exhibit the surface morphology of the Cu@Si

CSNWF after 30 cycles; the core–shell nanowires still preserve their shape and integrity with a diameter distribution ranging between 400 and 850 nm (Figure 6b) larger overall than those before cycling, as a result of the lithiation-induced volume expansion upon the cycled discharge–charge processes. Furthermore, after 500 cycles, the core–shell nanowire shape is hard to discern because the Cu@Si core–shell nanowires expand enough to make contact with each other and form a continuous film (Figure 6c). A more detailed analysis by cross-sectional STEM (Figure 6d) illustrates that the Si shell still adheres well to the Cu-core and the Cu nanowires remain as efficient electron transport pathways, securing a good rate capability of the Cu@Si CSNWF anode. In contrast, a large amount of cracks are observed on the surface of the pure Si film after 20 cycles, separating the Si film into 1–5 µm islands (Figure 6e) and resulting in poor cycling performance.

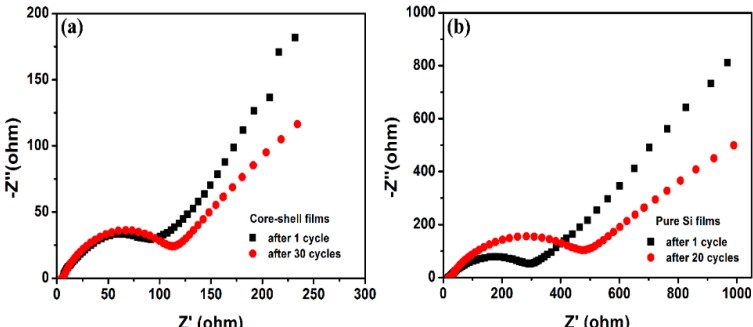

**Figure 5.** Electrochemical impedance spectroscopy spectra of (**a**) Cu@Si CSNWFs after 1 and 30 cycles; and (**b**) pure Si films after 1 and 20 cycles.

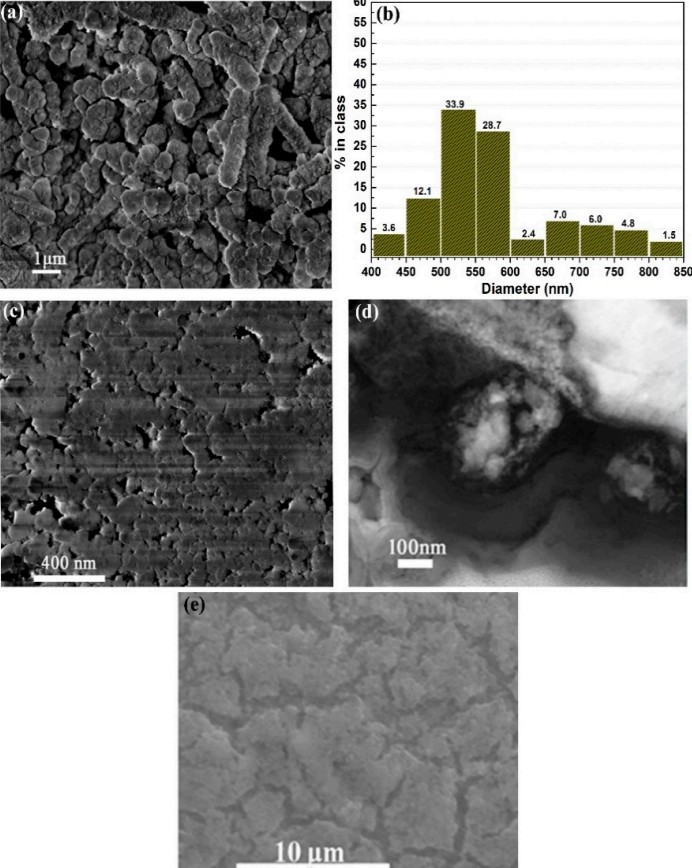

**Figure 6.** (**a**,**b**) SEM image and corresponding diameter distribution of the Cu@Si CSNWFs after 30 cycles; (**c**,**d**) SEM image and cross-section STEM image of the Cu@Si CSNWFs after 500 cycles; and (**e**) SEM image of the pure Si film after 20 cycles at a current rate of 0.1 C between 0.01 and 1.5 V.

## 4. Conclusions

Cu@Si CSNWFs were fabricated through slurry casting and subsequent magnetron sputtering and investigated as anode materials for lithium ion batteries. Both the silicon shell and copper nanowire core are in the crystal state and they are adhered by an interface layer of Cu3Si, which could enhance the Cu/Si interface strength; the Cu nanowire network functions well as an electron conductive path. These endow Cu@Si CSNWFs with a superior electrochemical performance with a high discharge capacity of about 948 mAh/g after 500 cycles, corresponding to a capacity fading rate of 0.2% per cycle. This work demonstrates that Cu@Si CSNWF is a promising anode material for high-energy-density lithium ion batteries.

**Author Contributions:** Formal analysis, Z.X.; methodology, J.Y.; writing—original draft, L.Z. (Lifeng Zhang); writing—review and editing, L.Z. (Linchao Zhang). All authors have read and agreed to the published version of the manuscript.

**Funding:** This work was supported by National Natural Science Foundation of China (Grant No. 51502300), The fund of Science and Technology on Reactor Fuel and Materials Laboratory (Grant No. 6142A06200310), Anhui Provincial Natural Science Foundation (Grant No. 1608085QE88) and Hefei Center of Materials Science and Technology (2014FXZY006).

**Institutional Review Board Statement:** Not applicable.

**Informed Consent Statement:** Not applicable

**Data Availability Statement:** No new data were created or analyzed in this study. Data sharing is not applicable to this article.

**Conflicts of Interest:** The authors declare no conflict of interest.

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
