# Peer review of "Cu&Si Core–Shell Nanowire Thin Film as High-Performance Anode Materials for Lithium Ion Batteries"

_applsci, doi:10.3390/app11104521_

Round 1
Reviewer 1 Report
Comment 1: You should write Cu3Si instead of Cu3Si. The "3" must be subscript.
Comment 2: In line 40, is it "a-Si" or "c-Si"?
Comment 3: In page 2 ("Materials and methods") specify the volume of isopropanol and quantity of CSNWFs used.
Comment 4: In line 85, "of the density" is repeated!
Comment 5: In concentration, mol.L-1, the "-1" must be superscript, as so as the degree sign in 280ºC.
Comment 6: It is missing (g) in the legend of Figure 2, please fix it.
Comment 7: In line 169, you write "specific capacity", should it not be "discharge capacity"?
Comment 8: Relatively to cyclic voltammetry, there is an inconsistence, because in the legend of Figure 3 you say that it is up to 1.2 V, but in the text you say 1.0 V. Please check it!
Reviewer 2 Report
In this work, Yang et al. developed a hybrid film based on Cu&Si core-shell nanowire and investigated its application as an anode material for lithium-ion batteries. The manuscript is well written and organized. The experiments are conducted carefully and the data are convincing. I suggest the acceptance of this paper after the following minor concerns are addressed.
- The first sentence in the Introduction is not necessary. Also, check the language of the whole manuscript thoroughly.
- The author claims that “In comparison with other core materials, Cu nanowire core has an obvious advantage such as higher electrical conductivity and fracture toughness”. However, I did not find any evidence about the comparison of fracture toughness among a variety of core materials. Is there any reference that provides such a kind of comparison? If yes, please cite the reference properly.
Reviewer 3 Report
The present paper reports an experiental study of the electrochemical properties of Cu-Si core-shell nanowire-thin film strutures used as anode material in Li-ion batteries. The authors find that the introduction of Cu nanowires leads to a stabilisation of the Si anode and to an improved performance with cycling. The paper is well written and the work seems to be well carried out, with a detailed structural and morphological characterisation of the Cu-Si core-shell nanowire-thin films before and after cycling the electrode. Voltamometry measurements indicate an improvement of the Li-ion battery performance for the composite anode as opposed to the bare Si film, with the authors suggesting that the mechanism behind the improvement is related to the mechanical stability provided by the hollow Cu nanowires and formation of a passivation interphase layer upon cycling. In my view, the work presents new results that are likely to be of interest to the applied sciences community and I can recommend it for publication in this journal.
